# Associations between Smoking Status and Health-Related Physical Fitness and Balance Ability among Older Males in Taiwan

**DOI:** 10.3390/medicina59071350

**Published:** 2023-07-23

**Authors:** Yi-Chuan Hung, Po-Fu Lee, Chi-Fang Lin, Yan-Jhu Su, Jenn-Woei Hsieh, Yu-Ju Lin, Chien-Chang Ho, Yun-Tsung Chen

**Affiliations:** 1Department of Sport Management, National Taiwan University of Sport, Taichung City 404, Taiwan; yichuan1206@gmail.com; 2Sports Administration, Ministry of Education, Taipei City 104, Taiwan; 3Department of Leisure Industry and Health Promotion, National Ilan University, Yilan County 260, Taiwan; pflee@niu.edu.tw; 4Department of Physical Education and Sport Sciences, National Taiwan Normal University, Taipei City 106, Taiwan; aa830521@gmail.com; 5Department of Gerontology, University of Massachusetts Boston, Boston, MA 02125, USA; yanjhu.su001@umb.edu; 6Department of Physical Education, Fu Jen Catholic University, New Taipei City 24205, Taiwan; 032508@mail.fju.edu.tw (J.-W.H.); 093703@mail.fju.edu.tw (C.-C.H.); 7Office of Physical Education, Fu Jen Catholic University, New Taipei City 24205, Taiwan; 8Department of Nursing, Yuanpei University of Medical Technology, Hsinchu City 30015, Taiwan; cosa0314@yahoo.com.tw; 9Research and Development Center for Physical Education, Health and Information Technology, College of Education, Fu Jen Catholic University, New Taipei City 24205, Taiwan; 10Sports Medicine Center, Fu Jen Catholic Hospital, New Taipei City 243, Taiwan; 11Department of Health and Leisure Management, Yuanpei University of Medical Technology, Hsinchu City 30015, Taiwan

**Keywords:** physical fitness, cigarette smoking, older adults, Taiwan

## Abstract

The primary aim of this study was to examine the relationships between smoking status and health-related physical fitness and balance ability in older males residing in Taiwan. This investigation adopted a cross-sectional design, utilizing data from 7688 older males who took part in the 2014–2015 wave of the National Physical Fitness Survey of Taiwan. Various data sources, including a standardized structured questionnaire, anthropometric measurements, health-related physical fitness assessments, and balance ability tests, were analyzed. The participants were divided into three categories based on their smoking habits: never smokers, former smokers, and current smokers. Multiple regression analyses were performed to evaluate the linear association between cigarette smoking status and health-related physical fitness and balance ability performance. Health-related physical fitness and balance performance were significantly greater (*p* < 0.05) in the never smoker group than in the current smoker group. Current smoking status was significantly negatively (*p* < 0.05) associated with cardiopulmonary function, muscular endurance, flexibility, and balance performance. A history of smoking (former smoker) was significantly negatively (*p* < 0.05) associated with the 2-min step test, 30-s arm curl and chair stand, as well as the 8-foot up-and-go test; however, the association was not significant for the back scratch, chair sit-and-reach, and one-leg stance with eyes open performance. These results suggest that current cigarette smoking is detrimental to health-related physical fitness and balance performance in older males. Quitting smoking may reverse the effects of smoking on overall body flexibility and static balance performance in Taiwanese older adults, thereby reducing the risk of falls and incapacity.

## 1. Introduction

Non-communicable diseases (NCDs) cause 41 million deaths per year and mainly comprise cardiovascular diseases (e.g., coronary artery disease and stroke), chronic respiratory diseases (e.g., asthma and chronic obstructive pulmonary disease), diabetes mellitus, and cancer [1]. It has been reported that smoking, physical inactivity, an unhealthy diet, and high alcohol intake are the major behavioral risk factors for NCDs [2].

Notably, smoking has been found to be the second leading risk factor for mortality globally [3]. It has been shown that smoking is associated with a 4.8-year reduction in life expectancy during the age bracket of 40–85 years, and a life expectancy reduction of 3.9 years for diabetes, 2.4 years for physical inactivity, and 1.6 years for hypertension [4]. In addition, smoking not only impacts health status but also influences health-related physical fitness [5], which may increase the risk of disability and reduce quality of life, especially in older adults. Previous studies have found that the adverse effects of smoking on cardiovascular fitness are reversible in middle-aged men [6,7]. However, the reversibility of smoking effects on health-related physical fitness for older adults remains unclear.

Health-related physical fitness consists of cardiopulmonary ability, muscular strength and endurance, flexibility, and body composition [8]. It has been found that smoking is correlated with decreased cardiopulmonary function and muscle endurance capacity, as evidenced in the 3000-m run, 2-min sit-ups, and 2-min push-ups tests [9,10]. However, compared with the 3000-m run, the 2-min step test is a safe, well-tolerated, time-efficient, and valid method for predicting cardiopulmonary function for older adults and adults with low fitness levels [11]. Moreover, a study reported that smokers have lower flexibility than non-smoking athletes [12]. Previous studies indicate that older adults with poor muscle strength, muscle endurance, or flexibility may have a reduction in balance and an increase in falling risk, which can result in fractures and even mortality [13,14]. Relatively limited information is available regarding the relationship between smoking status and balance in older populations.

Taiwan is an aged society, and approximately 17% of individuals were aged ≥ 65 years in 2021 [15]. In this highly technologically developed country, older males aged 66 years and older have a higher proportion of smoking behavior than older women (7.4% vs. 0.8%) [16]. However, to our knowledge, no study has yet reported that aging combined with smoking behavior may deteriorate health-related physical fitness performance when compared with older adults who have never smoked and former smokers. Therefore, through this study, we aimed to examine the relationships between cigarette smoking status and health-related physical fitness and balance ability performance in older males residing in Taiwan.

## 2. Materials and Methods

### 2.1. Study Design and Participants

The cross-sectional data used in this study were derived from Taiwan’s 2014–2015 National Physical Fitness Survey (TNPFS), which was conducted by the Sports Administration within the Ministry of Education in Taiwan. A convenient sampling strategy was used to recruit participants from 46 physical fitness test stations in 22 cities and counties in Taiwan between October 2014 and March 2015, and all participants were only calculated once [17]. For representative purposes, the participants were selected using multiple stratifications based on geographical area, age, and gender. The study incorporated a comprehensive survey methodology, which involved conducting face-to-face interviews utilizing a standardized structural questionnaire. Additionally, trained examiners and medical specialists, such as nurses and doctors, performed anthropometric measurements, health-related physical fitness assessments, and balance ability tests. The inclusion criteria for the participants of the present study were: (1) ≥65 years of age; (2) male; and (3) Taiwanese, and the exclusion criteria were: (1) individuals lacking information on health-related physical fitness and balance ability; (2) individuals with systolic blood pressure ≥ 140 mmHg and/or diastolic blood pressure ≥ 90 mmHg; and (3) individuals with heart disease, hypertension, chest pain, vertigo, and/or musculoskeletal disorders. With these selection criteria applied, a total of 7688 older Taiwanese men remained in our analyses. This study followed the Tenets of the Declaration of Helsinki, and ethics approval was obtained from the Institutional Review Board of Fu Jen Catholic University in Taiwan (FJU-IRB C110113).

### 2.2. Measurements

#### 2.2.1. Covariates

This study included sociodemographic characteristics (age, gender, educational level, monthly income, and marital status), cigarette smoking status, betel nut chewing habits, and self-perceived health as control variables. Education was categorized into three levels: elementary school or lower, junior or senior high school, and college or higher. Monthly income was categorized as ≤NTD 20,000, NTD 20,001–40,000, and ≥NTD 40,001. Marital status was represented by three categories: married, never married, and divorced/separated/widowed. Betel nut chewing habits were characterized into the following categories: “never users” (those who never established the habit), “former users” (those who had the habit but had quit), and “current users” (those who continued with the habit). Self-perceived health status included three categories: excellent or good, fair, and very bad or poor.

#### 2.2.2. Anthropometric Variables

Anthropometric variables, including body weight, height, waist circumference (WC), and hip circumference (HC), were assessed in this study. Body weight was measured with a precision of 0.1 kg using a scale, while the participants wore light underclothes. Height was measured with a precision of 0.1 cm using a wall-mounted metal measuring tape and an acute-angled headpiece. The participants stood against a plumb-checked vertical wall, with their shoes removed. Body mass index (BMI) was calculated as body weight (kg) divided by the square of the height (m^2^). Obesity was defined according to the cut-off points of BMI classification for the adult and elderly population in Taiwan, which was adopted as established by the Health Promotion Administration, Ministry of Health and Welfare in Taiwan; individuals were classified as underweight (BMI < 18.5 kg/m^2^), normal weight (18.5 ≤ BMI < 24 kg/m^2^), overweight (24 ≤ BMI < 27 kg/m^2^), or obese (BMI ≥ 27 kg/m^2^). WC (measured to the nearest 0.1 cm) was measured twice, midway between the lowest rib and iliac crest, following normal exhalation, and the mean value was used. Hip circumference (measured to the nearest 0.1 cm) was measured twice at the point of maximum protrusion of the buttocks below the hip bones, and the mean value was utilized. Waist-to-hip ratio (WHR) was calculated by dividing waist circumference (in cm) by hip circumference (in cm).

#### 2.2.3. Cigarette Smoking Status

The assessment of cigarette smoking status in this study was obtained through a self-report questionnaire administered during the interview. The participants were categorized into three groups based on their smoking habits: never smokers (individuals who had never initiated smoking), current smokers (individuals who were actively smoking at the time of the study), and former smokers (individuals who had previously smoked but had successfully quit).

#### 2.2.4. Health-Related Physical Fitness and Balance Ability

In order to evaluate the functional capacity of elderly males in Taiwan, this study assessed three key components of health-related physical fitness and two primary components of balance ability, using a total of seven measures. The physical fitness measures included cardiopulmonary endurance (2-min step), muscle strength and endurance (30-s arm curl and 30-s chair stand), and flexibility (back scratch and chair sit-and-reach). The balance ability measures consisted of a one-leg stance with eyes open and the 8-foot up-and-go test. Performance standards for each physical fitness and balance ability measurement were established for males and females aged 65 years and older, based on data derived from an annual national survey encompassing over 20,000 older adults in Taiwan. Trained examiners, who had completed an official training seminar and passed a certification test on standardized procedures, recorded the measurements and collected the aforementioned data. The methodologies and procedures employed for the 2-min step, 30-s arm curl, 30-s chair stand, back scratch, chair sit-and-reach, and 8-foot up-and-go tests strictly adhered to the guidelines presented in the Senior Fitness Test manual. Furthermore, the one-leg stance with eyes open test followed the protocols outlined in previous studies. The specific details pertaining to each measurement item are elaborated as follows:

(1) Aerobic endurance (2-min step test): For the assessment of aerobic endurance using the 2-min step test, a wall was marked with colored tape to indicate the midpoint between the patella and iliac spine. This marking served as a reference for the height to which participants were instructed to raise their knees while walking in place. The total number of steps completed by the participants within a 2-min duration was recorded.

(2) Muscle strength and endurance in the upper extremities (30 s arm curl): To assess muscle strength and endurance in the upper extremities using the 30-s arm curl test, the participants were positioned on the edge of a chair, specifically on the side corresponding to their dominant hand. They were instructed to maintain a straight back and ensure both feet were firmly planted on the floor. Holding a dumbbell in their dominant hand, the participants were then instructed to bend their elbow, bringing the weight towards their shoulder. The number of arm curls, which involved bending the elbow towards the shoulder and returning to the initial position, completed within a 30-s timeframe was recorded. 

(3) Muscle strength and endurance in the lower extremities (30 s chair stand test): To evaluate muscle strength and endurance in the lower extremities using the 30-s chair stand test, the participants were instructed to sit in the middle of a chair with their feet fully grounded on the floor. Their arms were crossed in front of their chest. The number of times each participant stood up from the seated position and returned to sitting within a duration of 30 s was recorded. 

(4) Flexibility in the upper extremities (back stretch tests): To assess flexibility in the upper extremities using the back stretch test, the participants were instructed to position their dominant hand behind their shoulder on the same side, with their palm facing their back. They were then asked to extend their other hand, with their palm facing away from their back, and reach upward from the lower back towards their dominant hand, attempting to overlap the two hands or stretch as far as possible. The test outcome was determined based on the distance between the middle fingers of the two hands. Participants received negative points if their hands did not touch, indicating a greater distance between the fingers, and positive points if there was any overlapping between the hands. 

(5) Flexibility in the lower extremities (chair sit-and-reach test): To evaluate flexibility in the lower extremities using the chair sit-and-reach test, the participants were instructed to bend one leg while keeping the other leg straight with the heel in contact with the floor. They were then asked to place their hands on top of each other and reach towards the toes of the extended leg, maintaining the position for 2 s. This process was repeated twice for each leg. The participants received negative points corresponding to the distance between their fingertips and the tips of their toes, and positive points were assigned for the distance their fingertips extended beyond their toes. The highest score achieved by each participant was recorded as the test result. 

(6) Dynamic balance ability (8-foot up-and-go test): To assess dynamic balance ability using the 8-foot up-and-go test, the participants were seated on a chair while a traffic cone was placed at a distance of 2.44 m in front of them. Upon the tester’s instruction of "start," the participants were required to walk, without running, as swiftly as possible around the traffic cone, return to the starting point, and then resume sitting on the chair. 

(7) Static balance ability (one-leg stance with eyes open test): For the evaluation of static balance ability using the one-leg stance with eyes open test, the participants were instructed to stand with their hands placed on their waist. They were then asked to raise one leg and position it on the inner side of the ankle of the opposite leg. This test was performed alternately for both legs. A perfect score of 120 s was awarded for maintaining the one-leg stance for the entire duration.

Following an explanation of the physical fitness and balance ability measurements, the participants were provided with a 10-min warm-up period to optimize their performance. The measurements were scheduled prior to the commencement of exercises. To ensure adequate rest between measures, all participants were evaluated in the following sequence: body weight, height, WC, HC, one-leg stance with eyes open, 30-s chair stand, 30-s arm curl, 2-min step, chair sit-and-reach, back scratch, and 8-foot up-and-go. A rest period of 3 to 5 min was allowed between each measurement.

### 2.3. Statistical Analyses

Statistical analysis was conducted using SAS version 9.4 (SAS Institute, Cary, NC, USA) software in this study. Categorical variables are expressed as percentages. Continuous variables are expressed as the means with their standard errors (SEs). Chi-square (*x*^2^) tests were used to compare differences between the groups by cigarette smoking status for categorical variables, and one-way analyses of variance (ANOVAs) were used for continuous variables. Multiple regression analyses were used to assess the linear associations of cigarette smoking status with health-related physical fitness measurements and balance ability after adjusting for age, general and abdominal obesity, educational level, monthly income, self-reported health status, and betel nut chewing habits. All statistical tests were two-tailed, and *p* < 0.05 was considered statistically significant.

## 3. Results

Demographic characteristics and anthropometric indices of the 7688 participants aged 65 years and older included in the analyses are shown in Table 1. Significant differences were identified between cigarette smoking status and all demographic characteristics except for body weight. According to Tukey’s post hoc test, participants in the former smokers group had a higher weight (67.01 ± 0.36 kg), BMI (24.97 ± 0.12 kg/m^2^), WC (90.61 ± 0.34 cm), HC (96.96 ± 0.24 cm), and WHR (0.93 ± 0.002) than the other groups (i.e., current smokers group and never smokers group).

Table 2 shows the comparison of the health-related physical fitness measurements between the cigarette smoking status groups. Significant differences among the three groups were found on most health-related physical fitness measurements, except for the chair sit-and-reach test (*p* = 0.0691). Participants in the never smokers group reported the highest scores in the 2-min step test (*p* < 0.0001), 30-s arm curl (*p* = 0.0021), 30-s chair stand (*p* < 0.0001), back scratch (*p* < 0.0001), 8-foot up-and-go (*p* < 0.0001), and one-leg stance with eyes open test (*p* = 0.0002) among all groups. Participants with a history of smoking had significantly lower levels than the participants who had never smoked in the 2-min step test, 30-s arm curl and chair stand, as well as the 8-foot up-and-go test.

Table 3 shows the results of the multiple regressions for the associations between cigarette smoking status and health-related physical fitness measurements after adjusting for potential confounders. When adjusting for age (Model 1), both the former smokers and the current smokers had significantly lower scores on the 2-min step test (*p* < 0.0001, *p* < 0.0001), 30-s arm curl (*p* = 0.0151, *p* < 0.0001), 30-s chair stand (*p* < 0.0001, *p* < 0.0001), back scratch (*p* = 0.0137, *p* < 0.0001), 8-foot up-and-go (*p* < 0.0001, *p* < 0.0001), and one-leg stance with eyes open test (*p* = 0.0174, *p* < 0.0001) than the reference group (never smokers). For the chair sit-and-reach test, the current smokers also had significantly lower scores than the never smokers (*p* = 0.0049), while the former smokers did not have significant results compared with the never smokers (*p* = 0.4040). After adjusting for general and abdominal obesity, educational levels, monthly income levels, self-reported health status, and chewing betel nuts (Model 2), current smokers had significantly lower scores on all health-related physical fitness measurements than the reference group (never smokers). However, compared with the never smokers, the former smokers no longer had statistically significant results on the back scratch and one-leg stance with eyes open tests in Model 2 (*p* = 0.3539, *p* = 0.9494, respectively).

## 4. Discussion

In this study, we analyzed the relationships between cigarette smoking status and health-related physical fitness and balance performance using data from 7688 older Taiwanese males. The main findings of this study were as follows: (1) current smoking was negatively associated with cardiopulmonary function, upper and lower extremity muscular endurance and flexibility, and static and dynamic balance performance, and (2) former smoking was negatively associated with cardiopulmonary function, upper- and lower-extremity muscular endurance, and dynamic balance performance. These findings suggest that quitting tobacco use may restore flexibility and static balance performance in older adults.

It has been reported that the highest proportion (48%) of smoking behavior was observed among middle-aged men between the ages of 45 and 54 years worldwide in 2020, and the second greatest prevalence (37%) of tobacco use occurred in older adults between the ages of 65 and 74 years [18]. In this study, 80% of older males never smoked, had a lower WHR, and had a higher proportion of self-reported excellent or good health status. Conversely, 11% of older males smoked, had a higher WHR (0.93 vs. 0.92), comprised a lower proportion of the college or higher educational level (17% vs. 25%), and had self-reported excellent or good health status (63% vs. 70%) when compared with those who have never smoked.

A previous study reported that smoking increases oxidative stress and reduces the expression of endothelial nitric oxide synthase in pulmonary arteries, which may impair mitochondrial function and decrease oxygen delivery capacity, thereby reducing the ability of mitochondria to generate adenosine triphosphate (ATP) and inducing muscle contractile dysfunction; this may result in cardiopulmonary function, muscle force, and muscle endurance performance reduction [19,20]. However, Jeon et al. found no significant differences in cardiopulmonary endurance performance (6-min walking test) between those with a smoking status and non-smoking status in Korean older males and females [5]. In contrast, we found that current smoking was negatively associated with 2-min step test performance. These disparate results may be the result of the differences in aerobic testing methodology. Therefore, further research is required to clarify the impact of smoking on cardiopulmonary function among older adults of different ethnicities.

Furthermore, we observed that current smoking was negatively associated with 30-s arm curl and 30-s chair stand performance in older males. A similar study showed that current smoking decreased the participants’ repetitions of sit-ups in 30–60 s and an almost significant reduction in 30-s sit-to-stand performance (16.8 repetitions vs. 19.6 repetitions, *p* > 0.05) when compared with middle-aged and older-aged men who never smoked, but no difference was observed in grip strength performance [5,21]. It has been reported that grip strength is a good predictor of overall body strength and has been shown to be associated with the risk of cardiovascular disease and metabolic syndrome as well as quality of life [22,23,24,25]. In addition, Wüst et al. found that no difference was observed in knee extension strength between smokers and nonsmokers [26]. Therefore, we suggest that smoking may reduce oxygen delivery to working muscles more and that it is thus detrimental to overall muscular endurance performance without affecting muscle strength [19,26].

Aging is strongly associated with a decrease in flexibility, resulting in dysfunction, incapacity, and deterioration in health status [27,28]. It has been shown that independent older adults of both genders have a significantly greater range of motion in shoulder and hip flexibility [29]. In this study, we found that current smoking was negatively associated with back scratch and chair sit-and-reach performance in older males. A similar study reported that current smokers tended to perform worse in sit-and-reach performance (2.8 cm vs. 6.2 cm, *p* > 0.05) than older males who have never smoked [5]. Further research is required to clarify the mechanism by which smoking deteriorates the flexibility of older adults.

The 8-foot up-and-go test is not only a measurement of dynamic balance, agility, and speed but also a good predictor of falling for older adults [30]. To the best of our knowledge, this is the first study to report that current smoking is negatively associated with 8-foot up-and-go and one-leg stance with eyes open performance in older adults. A similar study indicated that both genders of independent older adults have a significantly greater walking speed than dependent older adults (those in nursing or rest homes) [28]. The present findings suggest that older adults who smoke may attenuate speed, agility, and static and dynamic balance performance, thereby increasing the risk of falling and incapacity.

A previous study indicated that abstinence from smoking may improve aerobic performance, heart rate response, and oxygen saturation in young and middle-aged adults [6,7,31]. In this study, we found that the effects of smoking on the back scratch test, chair sit-and-reach, and one-leg stance with eyes open performance were reversible in former older smokers after adjusting for confounding factors (age, abdominal obesity, educational levels, self-reported health status, and chewing betel nuts). However, the adverse effects of smoking on cardiopulmonary function, muscular endurance, dynamic balance, and agility performance were not reversible in older adults. Therefore, quitting smoking may reverse the effects of smoking on overall body flexibility and static balance performance in Taiwanese older males.

The present study has some limitations. First, this study did not measure the muscle mass, blood gases (e.g., PO_2_, CO_2_, and O_2_ saturation), heart rate, or hormonal responses of the participants. Without these measurements, it may be difficult to comprehensively elucidate the mechanisms of how smoking affects health-related physical fitness and balance performance in older populations. Second, due to the cross-sectional nature of this study, we only suggest the reversibility of smoking effects rather than prove they are reversible.

## 5. Conclusions

In summary, this study demonstrated that current cigarette smoking is detrimental to cardiopulmonary function, muscular endurance, flexibility, and balance performance in older males. Quitting smoking may not only prevent chronic diseases but also ameliorate the effects of smoking on overall body flexibility and static balance performance, which may reduce the risk of falling and incapacity. Further longitudinal studies designed to clarify the effects of abstinence from cigarette smoking on health-related physical fitness and balance performance in older Taiwanese are warranted.

## Figures and Tables

**Table 1 medicina-59-01350-t001:** Characteristics of the study population according to cigarette smoking status among older male Taiwanese adults.

Variables	Cigarette Smoking Status			*p*-Value	Post Hoc
	Never (*n* = 6185)	Current (*n* = 856)	Former (*n* = 647)		
Age (years)	74.46 ± 0.09	73.47 ± 0.23	73.97 ± 0.26	0.0001 *	N > C
Height (cm)	163.36 ± 0.08	163.38 ± 0.21	163.96 ± 0.23	0.0532	
Body weight (kg)	65.62 ± 0.11	65.13 ± 0.33	67.01 ± 0.36	0.0002 *	N, C < F
BMI (kg/m^2^)	24.58 ± 0.04	24.37 ± 0.11	24.97 ± 0.12	0.0007 *	N, C < F
Obesity status (%)				0.0078 *	
Obese	20.36	21.50	24.11		
Overweight	36.20	31.31	33.54		
Normal weight	39.08	41.94	36.32		
Underweight	4.37	5.26	6.03		
WC (cm)	88.75 ± 0.11	89.16 ± 0.32	90.61 ± 0.34	<0.0001 *	N, C < F
HC (cm)	96.28 ± 0.08	95.64 ± 0.22	96.96 ± 0.24	0.0001 *	C < N < F
WHR	0.92 ± 0.001	0.93 ± 0.002	0.93 ± 0.002	<0.0001 *	N < C, F
Abdominal obesity (%)				<0.0001 *	
Yes	53.73	50.58	44.98		
No	46.27	49.42	55.02		
Educational level (%)				<0.0001 *	
Elementary school or lower	41.68	55.85	47.96		
Junior or senior school	33.86	27.45	29.47		
College or higher	24.46	16.71	22.57		
Income level (%)				0.0246 *	
≦NTD 20,000	78.82	82.38	76.02		
NTD 20,001–40,000	11.28	10.81	12.43		
≧NTD 40,001	9.90	6.81	11.55		
Marital status (%)				0.0561	
Never married	59.53	54.42	58.28		
Married	29.67	32.74	29.53		
Divorced/separated/widowed	10.80	12.84	12.19		
Self-reported health status (%)				<0.0001 *	
Excellent or good	70.06	62.81	58.14		
Fair	23.75	28.07	31.47		
Very bad or poor	6.18	9.12	10.39		
Chewing betel nut (%)				<0.0001 *	
Never	98.96	86.53	66.94		
Current	0.83	9.25	1.55		
Former	0.21	4.22	31.52		

Post hoc: Tukey’s post hoc test. Abbreviations: BMI, body mass index; C, current smokers; F, former smokers; HC, hip circumference; N, never smokers; NTD, New Taiwan Dollar; SE, standard error; WC, waist circumference; WHR, waist-to-hip ratio. Values are expressed as the means ± SE or percentages (%). * *p* < 0.05.

**Table 2 medicina-59-01350-t002:** Health-related physical fitness measurements according to cigarette smoking status among older Taiwanese male adults.

Variables	Cigarette Smoking Status			*p*-Value	Post Hoc
	Never (*n* = 6185)	Current (*n* = 856)	Former (*n* = 647)		
2-min step (step)	88.65 ± 0.30	81.39 ± 0.84	85.38 ± 0.97	<0.0001 *	N > F > C
30-s arm curl (rep)	17.85 ± 0.07	17.21 ± 0.19	17.38 ± 0.22	0.0021 *	N > C
30-s chair stand (rep)	15.38 ± 0.06	14.18 ± 0.16	14.61 ± 0.19	<0.0001 *	N > C, F
Back scratch (cm)	−10.79 ± 0.16	−13.26 ± 0.45	−11.93 ± 0.52	<0.0001 *	N > C
Chair sit-and-reach (cm)	2.65 ± 0.11	1.91 ± 0.29	2.44 ± 0.36	0.0691	
8-foot up-and-go (s)	7.09 ± 0.02	7.59 ± 0.07	7.40 ± 0.08	<0.0001 *	N < C, F
One-leg stance with eyes open (s)	16.02 ± 0.15	14.33 ± 0.38	15.33 ± 0.46	0.0002 *	N > C

Post hoc: Tukey’s post hoc test. Abbreviations: C, current smokers; F, former smokers; N, never smokers; SE, standard error. Values are expressed as the means ± SE. * *p* < 0.05.

**Table 3 medicina-59-01350-t003:** Multiple regressions for the associations between cigarette smoking status and health-related physical fitness measurements after adjustment for potential confounders.

Variables	Cigarette Smoking Status	Model 1 ^a^			Model 2 ^b^		
		*Β*	SE.	*p*-Value	*β*	SE.	*p*-Value
2-min step test (step)	Former	−3.64	0.96	<0.0001 *	−3.17	1.15	0.0060 *
	Current	−8.19	0.86	<0.0001 *	−5.91	0.92	<0.0001 *
	Never	1.00	—	—	1.00	—	—
30-s arm curl (rep)	Former	−0.56	0.23	0.0151 *	−0.59	0.28	0.0350 *
	Current	−0.84	0.20	<0.0001 *	−0.62	0.22	0.0046 *
	Never	1.00	—	—	1.00	—	—
30-s chair stand (rep)	Former	−0.89	0.19	<0.0001 *	−0.61	0.23	0.0076 *
	Current	−1.43	0.17	<0.0001 *	−0.98	0.18	<0.0001 *
	Never	1.00	—	—	1.00	—	—
Back scratch (cm)	Former	−1.29	0.52	0.0137 *	0.57	0.62	0.3539
	Current	−2.73	0.47	<0.0001 *	−1.89	0.49	0.0001 *
	Never	1.00	—	—	1.00	—	—
Chair sit-and-reach test (cm)	Former	−0.30	0.36	0.4040	−0.19	0.44	0.6588
	Current	−0.90	0.32	0.0049 *	−0.76	0.35	0.0300 *
	Never	1.00	—	—	1.00	—	—
8-foot up-and-go (s)	Former	0.37	0.07	<0.0001 *	0.33	0.09	0.0001 *
	Current	0.64	0.07	<0.0001 *	0.44	0.07	<0.0001 *
	Never	1.00	—	—	1.00	—	—
One-leg stance with eyes open (s)	Former	−1.03	0.43	0.0174 *	−0.03	0.51	0.9494
	Current	−2.38	0.38	<0.0001 *	−1.43	0.41	0.0004 *
	Never	1.00	—	—	1.00	—	—

Abbreviations: β, regression coefficient; SE, standard error. ^a^ Adjusted for age. ^b^ Adjusted for age, general and abdominal obesity, educational levels, monthly income levels, self-reported health status, and chewing betel nuts. * *p* < 0.05.

## Data Availability

The data that support the findings of this study are available from the Sports Cloud: Information and Application Research Center of Sports for All, Sports Administration, Ministry of Education in Taiwan, but restrictions apply to the availability of these data, which were used under license for the current study and as such are not publicly available. Data are however available from the authors upon reasonable request and with the permission of the Sports Cloud: Information and Application Research Center of Sports for All, Sports Administration, Ministry of Education in Taiwan.

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
