# Peer review of "Associations between Smoking Status and Health-Related Physical Fitness and Balance Ability among Older Males in Taiwan"

_medicina, 2023, doi:10.3390/medicina59071350_

Round 1

Reviewer 1 Report

Dear authors,

Thank your for your submission.

Please consider the minor revisions.

Good work!

Please, provide a moderate editing of English language

Author Response

For Reviewer #1:

Reviewer’s comments to author:

Q1: Minor editing of English language required.

A1: Thank you for the Reviewer’s suggestion. This manuscript has undergone English

language editing by MDPI. The editing certificate is attached at the end of this revised

manuscript.

Q2: I believe that the conclusion needs to be developed a little further to fully support all the conclusions that can be drawn from the study.

A2: Thank you for your suggestion. We have revised the conclusions (Page 8, Line 338-342).

Reviewer 2 Report

Dear authors, first of all thank you for your submission to Medicine. The authors explore a topic of extreme importance to society in general that is related to the scourge of smoking habits and its potential consequences for health and general well-being.

I think the authors have done an excellent job and deserve credit for it, the introduction is well structured and clearly states the study problem. The methods are correctly described and contain information that allows the replication of the procedures.

The results are correctly described as well as the discussion.

I believe that the conclusion needs to be developed a little further to fully support all the conclusions that can be drawn from the study.

Overall good job!

The text is well written.

Author Response

Thanks for your kind words.

Reviewer 3 Report

The current script (medicina-2376494) focused on to determine the association between cigarette smoking status and health-related physical fitness and balance ability performance among older males in Taiwan. It is worth mentioning that it was a research with large sample size, this is conducive to the expanded recognition population. I thought it appropriate for this manuscript to be published at this journal but the follow issues need to be addressed before the acceptance.

  1.  In Abstract, Taiwan’s National Physical Fitness Survey should be the one of the most important sources of data. So, it should be listed in the reference or in appendix.

2.  7,688 older males who participated in Taiwan’s National Physical Fitness Survey from 2014-2015. Is this the data for the participants in two years? And why did the authors pickup the data in two years? Are there participants in 2015 who have been recalculated?

3.  The age of participants does not seem to be so important in this research to determine the association between cigarette smoking status and health-related physical fitness and balance ability performance for older males. So, What is the difference between Model 1 and Model 2 b in Table 3. ?

4. What is the reason for being studied nearly 10 years in this research?

5. In section 3, the body mass index (BMI) should be BMI=weight/height squared. Please check the spelling.

6. What is that mean for the suggested reversibility of smoking effects in this research?

7. What is the author's opinion on health-related physical fitness and balance ability performance of middle aged male smokers? 

Author Response

Q1: Are all the cited references relevant to the research?

A1: Thank you for your suggestion. We have confirmed all the cited references relevant to the research in the manuscript.

Q2: In Abstract, Taiwan’s National Physical Fitness Survey should be the one of the most important sources of data. So, it should be listed in the reference or in appendix.

A2: Thank you for your suggestion. We have cited the reference of Taiwan’s National Physical Fitness Survey in the Materials and Methods section (Page 2, Line 90; [17]).

Q3: 7,688 older males who participated in Taiwan’s National Physical Fitness Survey from 2014-2015. Is this the data for the participants in two years? And why did the authors pick up the data in two years? Are there participants in 2015 who have been recalculated?

A3: Thank you for your suggestion. Our data collection period was from October 2014 to March 2015, and all participants only calculated once. For instance: A convenient sampling strategy was used to recruit participants from 46 physical fitness test stations in 22 cities and counties in Taiwan between October 2014 and March 2015, and all participants only calculated once (Page 2, Line 85-90).

Q4: The age of participants does not seem to be so important in this research to determine the association between cigarette smoking status and health-related physical fitness and balance ability performance for older males. So, What is the difference between Model 1 and Model 2 b in Table 3.?

A4: Thank you for your suggestion. It has shown that confounding factors (e.g., age, gender, self-reported health status, occupation, and education) may overestimate or underestimate the real magnitude of an association, or even change the direction of a real association (Arija et al., 2015). Therefore, the use of properly adjusted confounding factors is necessary to avoid bias and distortion.

In this study, Model 1 and Model 2 were represent the Adjusted for age and Adjusted for age, general and abdominal obesity et al., respectively (Page 7, Line 256-258)

Arija, V.; Abellana, R.; Ribot, B.; Ramón, J.M. Biases and adjustments in nutritional assessments from dietary questionnaires. Nutr. Hosp. 2015, 31, 113–118.

Q5: What is the reason for being studied nearly 10 years in this research?

A5: In Taiwan, the nationwide physical fitness testing of seniors started in 2014 and continues to this day. However, there was an item of smoking behavior in the face-to-face only in the wave of 2014-2015. Therefore, the data analyzed in this research is the only database in Taiwan that includes both smoking behavior and physical fitness of the older adults.

Q6: In section 3, the body mass index (BMI) should be BMI=weight/height squared. Please check the spelling.

A6: Thank you for your suggestion. We have checked the spelling and description (Page 3, Line 121-122).

Q7: What is that mean for the suggested reversibility of smoking effects in this research?

A7: Thank you for your suggestion. In this study, we found that current cigarette smoking is detrimental to cardiopulmonary function, muscular endurance, flexibility and balance performance in older males. However, quitting smoking may improve in overall body flexibility and static balance performance, which means the reversibility of smoking  side effects.

Q8: What is the author's opinion on health-related physical fitness and balance ability performance of middle aged male smokers?

A8: Thank you for your suggestion. In our opinion, Taiwan is an aged society, and aging may  decreases the health-related physical fitness and balance ability performance. However, current cigarette smoking elderly had lower cardiopulmonary function, muscular endurance, flexibility and balance performance when compared with never smoked elderly. In addition, we also found that quitting smoking may ameliorate the effects of smoking on overall body flexibility and static balance performance. Therefore, we suggested that quitting smoking may not only prevent the chronic diseases but also reducing the risk of falling and incapacity for older adults.